# The Prevalence of Abdominal Adiposity among Primary Health Care Physicians in Bahia, Brazil: An Epidemiological Study

**DOI:** 10.3390/ijerph18030957

**Published:** 2021-01-22

**Authors:** André Luiz Brandão Costa, Magno Conceição das Merces, Amália Ivine Costa Santana, Douglas de Souza e Silva, Rodrigo Fernandes Weyll Pimentel, Pedro Carlos Muniz de Figueiredo, Tatiana Santos Brandão, Julita Maria Freitas Coelho, Alex Almeida e Almeida, Kairo Silvestre Meneses Damasceno, Thais Regis Aranha Rossi, Marcio Costa de Souza, Iracema Lua, Dandara Almeida Reis da Silva, Monique Magnavita Borba da Fonseca Cerqueira, Antonio Marcos Tosoli Gomes, Jeane Freitas de Oliveira, Anderson Reis de Sousa, Thiago da Silva Santana, Maria Lúcia Silva Servo, Márcia Cristina Graça Marinho, Lucelia Batista Neves Cunha Magalhães, Arthur Pinto Silva, Sergio Correa Marques, Rafael Moura Coelho Pecly Wolter, Lucia Helena Penna, Luiz Carlos Moraes França, Ellen Marcia Peres, Pablo Luiz Santos Couto, Priscila Cristina da Silva Thiengo de Andrade, Livia Fajin de Mello dos Santos, Ana Victória Gomes Fonseca, Charles Souza Santos, Lívia Maria da Silva Gonçalves, Argemiro D’Oliveira Júnior

**Affiliations:** 1Department of Life Sciences, State University of Bahia (UNEB), Salvador, BA 41150-000, Brazil; andrelbcosta@hotmail.com (A.L.B.C.); rodrigo.pimentel@ebserh.gov.br (R.F.W.P.); kairodamasceno@hotmail.com (K.S.M.D.); thais.aranha@gmail.com (T.R.A.R.); mcsouzafisio@gmail.com (M.C.d.S.); daraareis@gmail.com (D.A.R.d.S.); moniquemagnavita@hotmail.com (M.M.B.d.F.C.); mcmarinho@uneb.br (M.C.G.M.); arthurps@outlook.com.br (A.P.S.); 2Health Sciences Postgraduate Program, School of Medicine, Federal University of Bahia (UFBA), Salvador, BA 40026-010, Brazil; amalia0807@gmail.com (A.I.C.S.); douglasss-gbi@hotmail.com (D.d.S.e.S.); liviajeje@yahoo.com.br (L.M.d.S.G.); argemiro@ufba.br (A.D.J.); 3University Hospital Complex Professor Edgard Santos (HUPES), Salvador, BA 40110-060, Brazil; pedro.figueiredo@ebserh.gov.br (P.C.M.d.F.); tatiana.brandao@ebserh.gov.br (T.S.B.); 4Federal Institute of Education of Bahia (IFBA), Simões Filho, BA 43700-000, Brazil; julitamaria@gmail.com; 5Transcend Clinic (TC), Salvador, BA 41730-101, Brazil; alexalmeida.a@gmail.com; 6Municipal Health Department, Salvador, BA 40010-010, Brazil; 7Department of Health, State University of Feira de Santana (UEFS), Feira de Santana, BA 44036-900, Brazil; ira_lua@hotmail.com (I.L.); tssantana@uefs.br (T.d.S.S.); luciaservo@yahoo.com.br (M.L.S.S.); 8School of Nursing, State University of Rio de Janeiro (UERJ), Rio de Janeiro, RJ 20551-030, Brazil; mtosoli@gmail.com (A.M.T.G.); sergiocmarques@uol.com.br (S.C.M.); luciapenna@terra.com.br (L.H.P.); lcmoraesfranca@hotmail.com (L.C.M.F.); ellenperesuerj@gmail.com (E.M.P.); profprithiengo@gmail.com (P.C.d.S.T.d.A.); liviafajin@gmail.com (L.F.d.M.d.S.); 9School of Nursing, Federal University of Bahia (EEUFBA), Salvador, BA 40110-060, Brazil; jeane.foliveira@outlook.com (J.F.d.O.); son.reis@hotmail.com (A.R.d.S.); 10Medicine School, University Center (UNIFTC), Salvador, BA 41741-590, Brazil; luceliamagalhaes@terra.com.br; 11Department of Social Psychology and Development, Federal University of Espírito Santo (UFES), Vitória, ES 29075-910, Brazil; rafaelpeclywolter@gmail.com; 12School of Nursing, FG University Center, Guanambi, BA 46430-000, Brazil; pabloluizsc@hotmail.com; 13School of Administration, Univértix College, Três Rios, RJ 25635-416, Brazil; gomesanavictoria3@gmail.com; 14Health Department, University of Southwest Bahia (UESB), Jequié, BA 45200-000, Brazil; charlesss@uesb.edu.br

**Keywords:** abdominal fat, health personnel, health care, body mass index, physicians

## Abstract

Background: Labor activities are demanding for workers and can induce occupational stress. Primary health care (PHC) workers have faced problems that can lead to the development of stress and abdominal obesity. The aim of this study was to estimate the prevalence of abdominal adiposity among primary health care physicians in the metropolitan mesoregion of Salvador, Bahia. Methods: This is a cross-sectional study conducted with physicians from the family health units (FHUs) of the metropolitan mesoregion of Salvador, Bahia, Brazil. The number of FHUs corresponded to 41 teams (52 physicians). Anamnesis was performed and a questionnaire was applied. The clinical examination consisted of measuring waist circumference (WC), blood pressure levels (BP), and body mass index (BMI), as well as examining for acanthosis nigricans. Blood samples were collected for biochemical dosages. The data obtained were analyzed by SPSS version 22.0. Results: The sample included 41 physicians (response rate: 78.8%), of which 18 were women (44.0%). The percentage of overweight participants represented by BMI was 31.7%. The hypertriglyceridemia prevalence was 29.2%. HDL-c was low in 48.7% of the participants. The waist circumference measurement revealed a prevalence of abdominal adiposity of 38.8% (women) and 34.8% (men). Conclusions: Medical professionals in PHC are more susceptible to having higher abdominal adiposity, especially female physicians.

## 1. Introduction

Obesity is a chronic disease characterized by an excessive accumulation of body fat, is capable of taking years off life expectancy according to its severity, and is evidenced by the body mass index (BMI) [1]. However, studies argue that BMI alone is not an appropriate indicator for estimating certain health risks. Researchers have determined that the visceral distribution of body fat can induce systemic inflammatory states, which, in turn, are capable of resulting in several deleterious health outcomes [2]. Abdominal adiposity is associated with an increased incidence of depression, type 2 diabetes mellitus, hypertension, coronary heart disease, dementia, various forms of cancer, and sudden death, regardless of BMI [3,4]. 

Currently, labor activities have increasingly demanded that the contemporary worker be more dynamic and engaged. Thus, this context imposed for fulfilling professional activities can induce occupational stress, which creates a risk scenario for diseases both at the psychological and physical levels [5]. In this scenario, labor represents a fine line between professional achievement and a pathological state. All this depends on how workers see themselves before their work and how they feel in the social context of which they are a part [6].

Primary health care (PHC) workers, such as physicians, have faced problems regarding their activities and demands at work. Among the obstacles, we can highlight requirements related to goals, insufficient resources, an incompatible number of workers for the development of certain activities, institutional obstacles, precarious work relationships, and coping with a diverse range of social vulnerabilities [7]. This conjuncture can lead to the development of stress and, consequently, abdominal obesity [7].

An Italian study conducted in 2012 analyzed the association between the prevalence of stress and metabolic syndrome among radiologists [8]. The authors observed a statistically significant association between metabolic syndrome in professionals with higher stress levels. Nevertheless, this study did not include a description of abdominal adiposity specifically in these professionals, even though this is one of the criteria for the diagnosis of this syndrome.

There are few studies that elucidate the prevalence of abdominal adiposity among physicians. One study conducted in the Kingdom of Saudi Arabia in 2011 showed a prevalence of abdominal adiposity in physicians of 26.2% [9]. Another study conducted in India in 2016 demonstrated a 17% increase in the abdominal circumference of the physicians studied [10]. Therefore, it is necessary to perform this study, since it can point out the effects of occupational stress among the professionals in question. Given this relevance, the aim of this study was to estimate the prevalence of abdominal adiposity among primary health care physicians in the metropolitan mesoregion of Salvador, Bahia.

## 2. Materials and Methods 

This is a cross-sectional epidemiological study conducted with physicians from the family health units (FHUs) of the metropolitan mesoregion of Salvador, Bahia, Brazil from September 2016 to January 2017. This research is linked to the multicenter project entitled “Burnout Syndrome and Metabolic Syndrome in Health Professionals of Primary Health Care in the State of Bahia”. The participating institutions are the State University of Bahia, the Medical School of the Federal University of Bahia, the State University of Feira de Santana, and the Nursing School of the State University of Rio de Janeiro.

The total number of FHUs, the locus of the study, corresponded to 41 teams, totaling 52 physicians. The eligibility criteria for the study were the following: all professionals that had a medical degree (MD) who were fully developing assistance activities and agreed to participate in the research by signing an informed consent form (ICF). Pregnant women; professionals who were undergoing drug treatment for obesity, depression, anxiety, or occupational stress; and professionals who worked in the administrative field were all excluded.

For data collection, anamnesis was performed, and a questionnaire was applied to collect sociodemographic, labor, lifestyle, and human biology information. The clinical examination consisted of measuring waist circumference (WC), blood pressure levels (BP), and body mass index (BMI), as well as examining for acanthosis nigricans. A blood sample was also collected for biochemical dosages. 

WC measurement was performed at the midpoint of the distance between the lower edge of the rib cage and the ilium, in the horizontal plane. The professionals surveyed were in an orthostatic position, with arms by their sides, feet together, weight divided between the legs, and gazing out towards the horizon. It was requested that the area to be measured was free of clothing for better measurement accuracy.

The following cut-off points were used: normal WC (<88 cm for women and <102 cm for men) and high-risk WC (≥88 cm for women and ≥102 for men).

Blood pressure was measured using a stethoscope (Littmann^®^, Classic III, 3M, Saint Paul, MN, USA) and an aneroid sphygmomanometer (BD^®^ adult medium size, Franklin Lakes, NJ, USA) that had been previously calibrated. Two measurements were made on the left upper limb of the nursing professional after five minutes of rest. The average value considered was taken between two measurements within 5 min after resting for five minutes.

Overweight (OW) was considered according to the body mass index (BMI) and classified according to the World Health Organization (WHO) guidelines: normal weight (BMI < 25.0 kg/m^2^), overweight (BMI ≥ 25.0 kg/m^2^ and < 30 25.0 kg/m^2^), and obesity (BMI ≥ 30.0 kg/m^2^). To identify the presence of acanthosis nigricans, the cervical region was evaluated, followed by the armpits; the flexor surfaces of the limbs; the periumbilical; and, finally, the inframammary region.

The data obtained were organized by using version 22.0 of the Statistical Package for the Social Sciences (SPSS, IBM, Armonk, NY, USA). Data analysis was performed using descriptive statistics (absolute and relative frequencies of all variables of interest, thus enabling prevalence estimates among the sample).

The study was approved by the Research Ethics Committee of UNEB (the Estate University of Bahia) under opinion no. 872.365/2014. 

## 3. Results

The study population corresponded to 41 physicians and a response rate of 78.8% was achieved. There were six refusals; one professional was on sick leave and four did not undergo biochemistry. The sociodemographic characteristics of the professionals showed a young population that was up to 35 years of age (73.1%), predominantly male (56.0%), non-black (83.0%), and with a partner (61.0%). Regarding the characteristics of the work, 87.9% of participants reported not taking refresher courses for their work, and 73.2% had graduated from medical school over 10 years ago. The employment relationship was temporary (78.1%), with compensation equal to or greater than three times the Brazilian minimum wage (100.0%). Most reported that there were rest breaks during the working day (85.3%). Out of the professionals studied, 12.1% reported having suffered some type of aggression at work. The majority of the workers were satisfied with their work (70.7%); however, they showed dissatisfaction with their economic situation (92.7%) (Table 1).

Regarding lifestyle characteristics, it was noted that most did not perform physical activities (70.8%), did not consume alcoholic beverages (58.6%), and did not smoke (83.0%). Excess weight represented by BMI reached 31.7%, with three (7.3%) individuals categorized as overweight and 10 (24.4%) as obese. The presence of hypertriglyceridemia was 29.2%. HDL-c was low, at 48.7%. The findings for the waist circumference measurement revealed a prevalence of abdominal adiposity of 38.8% among women and 34.8% among men (Table 2).

## 4. Discussion

To our knowledge, this is the first study to verify the prevalence of abdominal adiposity among PHC physicians in Brazil. The findings of the present study point to a considerable prevalence of abdominal adiposity—38.8% among women and 34.8% among men, with a general average of 36.6%. A study conducted among physicians in a hospital in the Kingdom of Saudi Arabia found a prevalence of approximately 26.2% [9]. This prevalence of abdominal adiposity differs from our findings. Certainly, Brazilian physicians working in PHC are more exposed to complex social determinants inherent to PHC—namely, violence, precarious temporary employment bonds, and low salaries [11].

In this study, the sociodemographic profile of PHC medical professionals showed a young population, mostly male and married, with an income equal to or greater than three times the Brazilian minimum wage and who did not self-identify as black, which corroborates the findings of the study conducted by Guarda et al., 2012, among family strategy physicians of the state of Pernambuco, Brazil [12]. Particularly, in relation to the gender variable, several studies point to a predominantly female insertion in PHC [13,14], evidencing the feminization of medicine, a trend observed mainly from the 1990s [15].

Concerning graduation time, it was noticed that, despite being young, the professionals reported periods since their graduations of greater than 10 years, in addition to post-graduation training. A similar finding was observed in a study conducted by Lima et al., 2016, with PHC workers in the city of Vitória, Espírito Santo, Brazil, which showed a population of young professionals who had attended from graduation courses to medical residency [16]. It is known that doctors from younger age groups nurture desires for academic qualification, as well as professional and social development. These points may explain the fact that a significant number of professionals are investing in their careers after graduation [17]. In addition, the labor market is increasingly competitive and demanding for professionals, and is no different for physicians [18]. Still considering that the professionals included in this study are younger, work in the literature conducted in Italy demonstrated that academic stress influences the appearance of psychological changes in health students. This reinforces the high prevalence of stress amongst this population, which could be related to the development of other pathologies associated with stress [19]. 

It was noticed here that a significant number of professionals reported not having specific training to work in PHC. The National Policy of Permanent Education in Health (Brazil) argues that the training of health professionals is critical for the quality of health actions and services provided to the population and is a qualification strategy for the management of health services and systems. When exercised in a meaningful and contextualized way, it ultimately works as a tool to equip the worker for PHC tasks. This is meaningful for the professional; contributes to better performance at work; and, thus, improves workers’ health [20].

It was evident during this study that the insertion of physicians in FHUs occurred under precarious temporary employment bonds. Studies state that, in the PHC context, this type of employment arrangement is mainly established among higher education professionals [21,22]. The predominance of precarious hiring among physicians working in PHC was evidenced by a study conducted in several Brazilian municipalities [23]. This fact jeopardizes the professional’s bond with the service and the population served, as this type of hiring is one of the main reasons for the high turnover of workers in FHUs. In addition, the ease of political use for granting these positions, allowed by this labor contract, can confirm a tradition of political party bargaining that is well recognized in smaller municipalities [24].

Admission via public admission tests and job stability were among the objectives outlined during the 14th National Health Conference. The final report of this conference identifies public admission as a means for employment in the SUS (the Brazilian Public Health System) to ensure professional development and improvement in working conditions [25]. In addition to being necessary, this conduct becomes an essential tool to make the fundamental principle of primary care policy possible, which is to develop bond relationships between the community and the teams.

Therefore, it is evident from this study that despite the proposals from the Ministry of Health to improve the employment conditions of professionals working in the health care field, there is still a certain precariousness in the labor ties established between some professional categories. In addition to interfering in the health care provided by these workers, this instability might also produce deleterious effects on the workers’ health [26].

High job satisfaction rates were observed among participants, which is a fact supported by the literature. Studies show that physicians working in PHC are more satisfied than those who work in hospitals [27]. In this aspect, despite the adverse context, one feels pleasure in performing and experiencing this kind of work, mainly because there is an identification with the proposal and recognition of the work. Job satisfaction is undoubtedly an element of positive health consequences [28]. A study conducted by Lima et al., 2014, among professionals in the Family Health Strategy of Rio Grande do Sul, Brazil, showed that creating a healthy work environment and a spirit of team working led to greater satisfaction among medical professionals [29]. In turn, the reasons for dissatisfaction listed were a lack of specific training to work in PHC, the high demand of users, and instances in which patients do not commit to their treatment. 

It was noticed that most professionals reported not performing physical activities on a regular basis, which is divergent from a study conducted with PHC workers in Bahia [30]. Conversely in that study, the professionals investigated were not overweight and had adequate blood pressure, fasting glycemia, and triglycerides levels.

This finding may lie in the fact that the participants included are a young population, and are therefore less exposed to stress. The literature points out that these conditions (abdominal obesity, elevated blood pressure, low serum HDL-c levels, elevated serum triglycerides, and impaired fasting blood glucose) are not only associated but also have a direct relationship with age [31]. In this sense, the practice of regular physical activity should be stimulated in this population as it contributes to physical and mental health and quality of life. When physical activities are rarely performed, this is associated with a higher prevalence of chronic non-communicable diseases, in addition to other health issues [32].

This study also observed a low prevalence of overweight in the individuals studied. This could be associated with the fact that the population exclusively consisted of physicians, who necessarily have a higher level of education, which is inversely related to the occurrence of becoming overweight. Evidence suggests that classes with a higher level of education have a lower number of obese individuals [33,34].

Abdominal obesity had a high prevalence in this study and may be related to poor physical activity among physicians. Regular physical activity is related to increased energy demand and, consequently, to greater caloric burning, leading to lower abdominal adiposity [35]. Therefore, the overlap between a sedentary lifestyle, poor food quality, and academic stress could generate a triad that would lead to abdominal adiposity in these professionals. This was also evidenced in a study carried out by Mota et al., 2012, who, researching the living conditions of medical students and residents, observed similar data about the high prevalence of abdominal adiposity among those who had a higher academic qualification and were close to the end of their graduation [36].

Considering that most of the professionals studied in our work had graduated more than 10 years ago, we can deduce that a longer time since graduation is related to greater abdominal adiposity. Professional responsibilities, study and work routines, and stress levels among physicians seem to have a cumulative effect on the increase in abdominal circumference [36].

An Italian study conducted in 2012 with radiologists reinforces the findings observed in the present study [8]. The authors demonstrated that there is indeed a positive relationship between the prevalence of stress in these professionals and the presence of metabolic syndrome (a clinical condition that considers abdominal adiposity as a diagnostic criterion). The study also related the increase in demand, effort, overcommitment, work tension; an imbalance between effort and reward; and a reduction in support or rewards to more frequent signs of metabolic syndrome.

In this study, it was evident that abdominal adiposity was present in a higher percentage among women, which may have a direct relationship with the lack of physical activity among female medical professionals, since the practice of physical exercises is linked to better physical and health conditions [37]. A study conducted by Silva, Sandre-Pereira, and Salles-Costa, 2011, concluded that physical inactivity was more prevalent among women due to double working hours, domestic service, lower monetary income, and greater limitations in public spaces [38]. These standards do not match the medical standards of the female physicians in this study, since all of them had incomes higher than three times the minimum wage and limited domestic responsibilities.

A study conducted in Italy in 2020 evaluated the presence of occupational stress and mental health in anesthesiologists and showed that female professionals had higher scores for anxiety and depression than their male colleagues, although the differences did not reach statistical significance [39]. Women in the Italian study also reported a perception that they exerted greater effort than men, felt less rewarded, and felt a stronger sense of injustice, in addition to having a lower sleep quality. This reinforces that the data found in the present study regarding the higher prevalence of abdominal adiposity in this population could be the result of stress interfering in the occurrence of abdominal adiposity. 

Hormonal imbalance has also been related to the occurrence of abdominal adiposity among women, so a greater endocrine oscillation may imply a greater probability of developing it [40,41]. Despite the consideration of hormonal imbalance, the literature points out that ovarian hormones exert metabolic function on adipose tissue, indicating that the physiological difference of sex can interfere with the occurrence of abdominal adiposity [42]. Given the above, it is plausible to highlight the importance of longitudinal studies to assess hormonal imbalance and the defects of acanthosis nigricans and obesity.

In the study, there were some limitations. The main limitation was the cross-sectional nature of the study, which prevents inferences being made about the sequence of events and does not address the causal relationship between the variables studied. Other limitations refer to the short evaluation period and the small sample size. Despite this, this study was able to evaluate the totality of the family health units of the delimited region, bringing an interesting view of the presence of abdominal adiposity amongst this population of physicians. Longer longitudinal studies may provide a better knowledge of the relationships between work factors and the occurrence of abdominal adiposity among primary care physicians.

## 5. Conclusions

The study points to a prevalence of abdominal adiposity of 36.6%, with a higher percentage among women physicians in our studied sample. An important finding in this study was the evidence of a lack of professional training about medical care support in most of the physicians surveyed, which reveals the poor quality of PHC services. It should be considered that the training of professionals improves the quality of care and working conditions. In addition, it was evident that most of the physicians in our sample did not have employment bonds or stability in their positions, which goes against the rules of public admission and worsens the relationships of these professionals with the work performed.

There are few articles available in the literature that address the subject similar to that analyzed in this study. Given the importance of the topic, more studies are needed so that we can understand more clearly how the working relationships of medical professionals can affect a greater prevalence of abdominal adiposity. Possibly, public policies to provide better health and work conditions for these professionals could be needed to mitigate the occurrence of body changes that reflect deleterious disorders in their health.

## Figures and Tables

**Table 1 ijerph-18-00957-t001:** Sociodemographic, economic, and labor characteristics of primary health care physicians. Mesoregion of Salvador, Bahia, Brazil, 2017 (*n* = 41).

Variables	*n*	%
Age		
Up to 35 years	30	73.1
Older than 35 years	11	26.9
Sex		
Female	18	44.0
Male	23	56.0
Race/color		
Black	07	17.0
Non-black	34	83.0
Years of graduation		
Up to 10 years	11	26.8
Greater than 10 years	30	73.2
Degree of education		
Graduation	19	46.3
Post-graduation (specialization)	22	53.7
Refreshing course for PHC activities		
Yes	05	12.1
No	36	87.9
Marital status		
Without a partner	16	39.0
With a partner	25	61.0
Domestic service		
Yes	04	9.7
No	37	90.3
Household income ^$^		
Up to 2 minimum wages	00	0
Equal to or greater than 3 minimum wages	41	100
Satisfaction with economic situation		
Yes	03	7.3
No	38	92.7
Type of Work employment		
Permanent	09	21.9
Temporary	32	78.1
Job satisfaction		
Yes	29	70.7
No	12	29.3
Aggression at work		
Yes	05	12.1
No	36	87.9
Rest break		
Yes	35	85.3
No	06	14.7

^$^ Minimum wage in 2017: R$ 937.00.

**Table 2 ijerph-18-00957-t002:** Lifestyle characteristics and human biology of primary health care physicians. Mesoregion of Salvador, Bahia, Brazil, 2017 (*n* = 41).

Variables	*n*	%
Do physical exercises		
Yes	12	29.2
No	29	70.8
Smokers		
Yes	07	17.0
No	34	83.0
Ingest alcoholic beverages		
Yes	17	41.4
No	24	58.6
Presence of *Acanthosis nigricans*		
Yes	03	7.3
No	38	92.7
BMI (kg/m^2^)		
With overweight	03	7.3
With obesity	10	24.4
Without overweight	28	68.3
Systolic blood pressure (mmHg)		
≥130	07	17.0
<130	34	83.0
Diastolic blood pressure (mmHg)		
≥85	06	14.6
<85	35	85.4
Fasting blood glucose (mg/dL)		
≥110	08	19.5
<110	33	80.5
Triglycerides (mg/dL)		
≥150	12	29.2
<150	29	70.8
HDL cholesterol (mg/dL)		
<50	20	48.7
≥50	21	51.3
**Waist circumference** (**cm** Female physicians)		
≥88	7	38.8
<88	11	61.2
Male physicians		
≥102	8	34.8
<102	15	65.2

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
