# Peer review of "The Prevalence of Abdominal Adiposity among Primary Health Care Physicians in Bahia, Brazil: An Epidemiological Study"

_ijerph, 2021, doi:10.3390/ijerph18030957_

Round 1

Reviewer 1 Report

The study elucidates the prevalence of abdominal adiposity among physicians. This study is interesting because it can point out the effects of occupational stress among the professionals in question. This study is useful because it can provide a better understanding of how working conditions and relationships of medical professionals can affect a higher prevalence of abdominal adiposity and inform public policies that better health and work conditions for these professionals could be provided.

 The discussion is relevant, interesting and well argued. The study concludes that medical professionals working in PHC are more susceptible to having higher abdominal adiposity, especially the female physicians. Study also highlights the lack of professional training or stability in the position of PHC physicians. The overlap between several factors explained in the discussion generates stress that will lead to abdominal adiposity in these professionals.

However, regarding the scientific data, the research design needs to be improved, if possible, in particular the characteristics of the lifestyles and the human biology of Primary Health Care physicians (Table 2). Some suggestions :

Data on weight gain and BMI changes between day 0 of the study (in September 2016) and the last day (in January 2017) could be provided to reinforce the results and discussion. Also, asking the physicians about their weight gain and BMI changes since the first day in their position.

Author Response

Dear reviewer,

We are grateful for the considerations provided. Unfortunately, with regard to the requests to complement the data of lifestyle characteristics, as the study was conducted 04 years ago, we do not have the suggested complementary data, which reveals a limitation of our work.

We are willing to perform other complements at the suggestion of the dear reviewer. Grateful,

The authors

Reviewer 2 Report

This descriptive study can be of interest to indicate the health conditions of a sample of Brazilian doctors. Some improvements need to be made in the manuscript.

it is surprising that 35 authors conducted a study on 41 people. We may wonder what the role of each of the authors has been and whether all have actually played a sufficient role to determine ownership of the work, or whether they should rather not be indicated in the acknowledgements.

The study collected subjective data on lifestyle habits, objective data on abdominal obesity and blood chemistry findings. The authors have not tried to make a correlation between these variables and this is not justifiable.

In the abstract, the authors should specify the characteristics of the sample (survey site, the number of people contacted, number of respondents).

The bibliography needs to be enriched, with reference not only to Brazilian studies. For example, in paragraph lines 87-92 they talk about the stress of doctors and cite an article on nurses. Rather, it is advisable to cite studies showing the presence of occupational stress in doctors [e.g.: Magnavita N, Tripepi G, Di Prinzio RR. Symptoms in Health Care Workers during the COVID-19 Epidemic. A Cross-Sectional Survey. Int J Environ Res Public Health. 2020 Jul 20;17(14):5218. doi: 10.3390/ijerph17145218.].

Hence, they should cite articles indicating the association between stress and abdominal obesity in doctors [Magnavita N, Fileni A. Work stress and metabolic syndrome in radiologists: first evidence. Radiol Med. 2014 Feb;119(2):142-8. doi: 10.1007/s11547-013-0329-0.

In the discussion, when dealing with problems in young workers (lines 172-180), they should compare their results with those of the literature, which indicate the presence of stress in health students [eg.: Guidi L, et al. Neuropeptide Y plasma levels and immunological changes during academic stress. Neuropsychobiology 1999, 40: 188-95]. A high level of stress in young doctors and female workers has been demonstrated in doctors at the forefront of the Covid-19 pandemic [Magnavita N, Soave PM, Ricciardi W, Antonelli M. Occupational stress and mental health of anaesthetists during the COVID-19 pandemic. Int J Environ Res Public Health 2020, 17, 8245; doi:10.3390/ijerph17218245].

The article is completely missing a Limitations section. Among the many limitations of this study, the authors must include and adequately discuss: the small number of participants, the absence of a control group

Author Response

Dear reviewer,

We are grateful for the considerations provided.

As reported in our methodology, this study is part of a larger project, with other analyses and interventions, which involves analysis of participants in several areas of a state of Brazil, large in extension, and with difficulties of locomotion between the areas. So this reinforces the need of many associated researchers for the execution of the project. This justifies the large number of authors of the study, where all had important participation to qualify them with authors of the study. The function performed by each author was mentioned in the manuscript sent to this journal.

This study aims to collect descriptive data on the prevalence of abdominal adiposity among primary care physicians. We did not have the purpose of performing correlations between the variables currently. Therefore, this analysis was not performed and presented in this manuscript and thus we justify the absence of this analysis.

The proposed changes to the abstract were made according to the suggestion of the valued reviewer.

The recommended bibliography was added to our study with the objective of enriching the introduction and discussion of the results, as recommended by the reviewer.

A limitations section was added in discussion.

Grateful,

The authors

Round 2

Reviewer 1 Report

The authors have revised the introduction and the discussion which is better argued ; references have been added (ref. 7, 10 for the introduction and ref. 18, 39 for the discussion). The article is good supported by literature reported in the bibliography. In comparaison to other published articles, this study, although it has some limitations (see report 1) and could be even stronger, is publishable due to an interesting introduction/discussion.

I do not have any further comments.

Author Response

Dear, thank you for your careful evaluation.
No further changes were requested to our manuscript.
Sincerely;

Reviewer 2 Report

The authors improved the manuscript, at the best of their possibilities

Author Response

(The authors gave the same response as above.)
